# Arbuscular Mycorrhizal Fungi Diversity in *Sophora japonica* Rhizosphere at Different Altitudes and Lithologies

**DOI:** 10.3390/jof10050340

**Published:** 2024-05-08

**Authors:** Limin Yu, Zhongfeng Zhang, Peiyuan Liu, Longwu Zhou, Shuhui Tan, Shitou Kuang

**Affiliations:** 1Guangxi Institute of Botany, Guangxi Zhuang Autonomous Region and Chinese Academy of Sciences, Guilin 541006, China; yulimin1998@163.com (L.Y.); zlw@gxib.cn (L.Z.); tsh@gxib.cn (S.T.); 2College of Life Sciences, Guangxi Normal University, Guilin 541006, China; 3School of Pharmacy, Guilin Medical University, Guilin 541006, China; 4Agriculture and Rural Affairs Bureau of Quanzhou County, Guilin 541599, China; aa15878373535@163.com

**Keywords:** *Sophora japonica*, arbuscular mycorrhizal fungi, lithology, altitude, soil properties

## Abstract

Arbuscular mycorrhizal fungi play a key role in mediating soil–plant relationships within karst ecosystems. *Sophora japonica*, a medicinal plant with anti-inflammatory and antitumor properties, is widely cultivated in karst areas of Guangxi, China. We considered limestone, dolomite, and sandstone at altitudes ranging from 100 to 800 m and employed Illumina sequencing to evaluate AMF diversity and identify the factors driving *S. japonica* rhizosphere AMF community changes. We showed that the increase in altitude increased *S. japonica* AMF colonization and the Shannon index. The colonization of limestone plots was higher than that of other lithology. In total, 3,096,236 sequences and 5767 OTUs were identified in *S. japonica* rhizosphere soil. Among these, 270 OTUs were defined at the genus level and divided into 7 genera and 35 species. Moreover, available nitrogen, soil organic matter, and available calcium content had a coupling effect and positive influence on AMF colonization and Shannon and Chao1 indices. Conversely, available phosphorus, available potassium, and available magnesium negatively affected AMF Shannon and Chao1 indices. Lithology, altitude, pH, and available phosphorus are important factors that affect the dynamics of AMF in the *S. japonica* rhizosphere.

## 1. Introduction

Arbuscular mycorrhizal fungi (AMF), a group of ancient fungi, can form stable symbiotic relationships with 80% of terrestrial plants; this phylum is composed of 345 morphological species and 338 virtual taxa (VT) from the SSU dataset, 1033 species hypothesis (SH) from the ITS dataset, and 58 taxa from the LSU dataset molecularly defined AMF species [1]. AMF can survive in various habitats with different geological landforms and hydrothermal conditions, including, among others, high mountains, farmlands, plains, and saline-alkali lands [2,3]. AMF can form symbiotic relationships with various plants such as *Pogostemon cablin* (Blanco) Benth., *Atractylodes lancea* (Thunb.) DC., *Fritillaria taipaiensis* P. Y. Li, and *Salvia miltiorrhiza* Bunge [3]. In these associations, AMF facilitate the exchange of carbon, nitrogen, phosphorus, and other elements with plants, thereby promoting plant growth. Moreover, this relationship with AMF has significant advantages and potential for improving the quality of secondary metabolites, improving soil structure, and enhancing plant drought and disease resistance [2,3,4]. Therefore, exploring the relationship between AMF diversity and ecological factors in the rhizosphere of *S.japonica* will provide references for understanding the intricate dynamics between plants and their fungal partners and their response to the environment in the future.

AMF colonization affects the nutrient absorption of plants; however, changes and colonization dynamics of AMF are limited by soil fertility, temperature, and humidity [5]. Soil fertility is fundamentally limited and directly affected by the underlying mother rock type, which further influences the AMF community [6]. In karst regions, various types of rocks, including limestone [7], clasolite [7], and dolomite [8], give rise to diverse soil types and microhabitats [8]. The physicochemical properties in different soil types have significant impacts on plant growth and development, AMF colonization capabilities, species diversity, and distribution [8,9]. Various trends, including increases [10,11], decreases [12,13], and cubic function curves [14] in AMF diversity with elevation have been observed in previous studies. These findings strongly confirm the recognition that microbial distribution and diversity are restricted by environmental factors. External factors, such as different altitudes and soil characteristics, have far-reaching effects on AMF diversity and colonization. Moreover, these properties are important influencing factors and predictors of AMF diversity changes in the plant rhizosphere [8,11]. Therefore, investigating the correlation between altitude, soil properties, and other ecological variables, alongside understanding how plant rhizosphere AMF communities respond to these environmental factors, may enhance the ability to predict the dynamic changes and ecological roles of AMF under different site conditions.

*Sophora japonica* cv. *Jinhuai* is a medicinal plant belonging to the *Sophora* genus in the Leguminosae family. Notably, this plant possesses anti-inflammatory and antitumor properties; therefore, it is widely used in medicine, healthcare, functional foods, and various industries [15]. The primary cultivation areas for *S. japonica* are located in Guangxi and Hunan, China. Within these regions, Quanzhou County, Guilin City, and Guangxi are considered particularly rich in cultivated *S. japonica*. However, at present, there remain some key problems associated with the cultivation of *S. japonica*, including limited plant growth and varying contents of active substances. Specifically, these limitations may be related to the different geological backgrounds of karst regions and the diversity of beneficial microorganisms within these soils. The soils in the Guilin and Guanxi regions of China are developed from varying mother rock, including limestone, dolomite, and sandstone. However, the effects of the mother rock and altitude on the *S. japonica* rhizosphere AMF community remain unclear. Therefore, evaluating the correlation between lithology, altitude, soil properties, and the dynamic changes in the AMF community and mycorrhizal colonization may be important in elucidating the ecological advantages of AMF in karst farmland.

Illumina sequencing technology was used to detect the molecular diversity of AMF in the rhizosphere of *S. japonica* under different altitudes and lithology treatments and to evaluate the driving factors of the AMF community. The purpose of this study was to explore the AMF symbiotic relationship and germplasm resources of medicinal plant *S. japonica*, and to provide basic data for understanding the symbiotic relationship of medicinal plant fungal partners and the application of AMF in medicinal plant cultivation.

## 2. Materials and Methods

### 2.1. Experiment and Sampling Method

This study was conducted at various locations in Dongshan Township, Jiantang Town, Quanzhou Town, Baibao Township, Yanshan District, Quanzhou County, Guilin City, and Guangxi Province, China. Quanzhou County (25°29′–26°23′ N, 110°37′–111°29′ E, 188 m altitude on average) has a subtropical monsoon climate [16] with an average annual temperature of 17.9 °C, an average annual rainfall of 1563.1 mm, and an average frost-free period of 294.6 d.

We considered the lithology and altitude of each plot, assessing three lithologies (sandstone, dolomite, and limestone) and seven altitude-gradient conditions. The date of planting was March 2017. The sampling date was June 2022. The sandstone (SY) sampling points were located in Yanshan and Dongshan, with 2 elevation gradients of 100–200 m and 600–700 m, represented by SY1 and SY2, respectively. The dolomite (BY) sampling points were located in Jiantang and Quanzhou towns, with 3 elevation gradients of 100–200, 200–300, and 200–300 m, represented by BY1, BY2, and BY3, respectively. The limestone (SH) sampling points were located in Baibao and Dongshan, with 5 altitude gradients of 300–400, 400–500, 500–600, 600–700, and 700–800 m, represented by SH1, SH2, SH3, SH4, and SH5, respectively.

We selected representative 5-year-old *S. japonica* plants that exhibited robust growth, consistent condition, and no signs of pests or diseases. Stones and debris on the soil surface were removed during sampling. Then, alcohol-disinfected clean shovels were used to dig beside the surrounding host plants. The roots of *S. japonica* were excavated as completely as possible while preserving the slender and fresh roots. A total of 2 kg soil samples were randomly collected in the 0~20 cm soil layer according to the S-shaped route. Three replicate plots were set for each plot. The sample of each replicate plot was a biological composite sample formed by a random combination of three root and rhizosphere soil samples. The distance between each plant was 2.5–3 m. These samples were transported to the laboratory in an icebox. Fresh fibrous roots were washed with ddH_2_O and soaked in a formalin–aceto–alcohol (FAA) fixative (CH_2_O (130 mL) + CH_3_COOH (50 mL) + 50% C_2_H_5_OH (2000 mL)) for subsequent mycorrhizal morphological determination. Plant rhizosphere soil refers to the interface between plant roots, microbial activities, and the external environment [17]. The plant roots were cleaned with a brush before collecting the *S. japonica* rhizosphere soil, which was subsequently divided into two parts. One portion was used for subsequent DNA extraction and high-throughput sequencing. The remaining samples were homogenized after air-drying at 20–25 °C; this allowed for AMF spores determination and the identification of soil physicochemical properties.

### 2.2. Determination of the S. japonica Soil Physicochemical Properties

A total of 10 mL of distilled water was added to 2 g soil sieved through a sieve with a diameter of 0.149 mm. After shaking and standing, the pH value was measured by a corrected pH meter (FE28, Mettler Toledo, Shanghai, China). The soil was added with 0.8 mol/L K_2_Cr_2_O_7_ and H_2_SO_4_, boiled at 180 °C for 5 min in an oil bath (HH-S, Jiangsu Kexi Instrument Factory, Changzhou, China), and then titrated with 0.2 mol/L FeSO_4_ [18]. The soil organic matter (SOM) content was calculated according to the oxidation correction coefficient and FeSO_4_ volume [18]. After adding MgCl_2_ to the soil, the content of available nitrogen (AN) in the soil was determined with an automatic Kjeldahl nitrogen analyzer (K1160, Shandong Hanonscientific Instruments, Dezhou, China) [19]. HCl/H_2_SO_4_ was added to the soil sample, and the indicator and display reagent were added to the supernatant. The absorbance was measured using an ultraviolet (UV)–visible spectrophotometer (UV-1800PC, Shanghai Mapada Instruments, Shanghai, China) [20]. The available phosphorus (AP) content was calculated according to the constant volume during color development [20]. Next, the air-dried soil samples were added to a 1 mol/L NH_4_OAc solution, filtered after shock, and analyzed using a flame photometer (FP640, Shanghai Jingke Electronics, Shanghai, China); the corresponding standard curve was used to calculate the soil available potassium (AK) content [21]. Finally, air-dried soil was added to a 1 mol/L CH_3_COONH_4_ solution. After shaking and centrifugation, the filtrate was collected, and the available calcium (ACa) and magnesium (AMg) content were determined using inductively coupled plasma optical emission spectrometry (ICAP-7200, Thermo Fisher Scientific, Waltham, MA, USA).

### 2.3. Determination of S. japonica AMF Colonization and Spore Density

Root segments were cut to 1 cm and heated at 95 °C for 1.5 h in a 20% (*w*/*v*) KOH water bath. Roots were decolored and softened with 10% H_2_O_2_ and acidified with 2% HCl. Dyeing was carried out with 0.05% triphenyl blue staining solution. The root segments were stored in a lactic acid glycerol solution (C_3_H_6_O_3_ (50 mL) + C_3_H_8_O_3_ (100 mL) + H_2_O (50 mL)) [22]. The presence of AMF hyphae, arbusculues, and vesicles in roots indicated AMF colonization in the plant. A total of 30 randomly selected root segments from each sample were observed and detected under a 20× biological microscope (Leica, Wetzlar, Germany). Mycorrhizal colonization (%) = number of root segments with mycorrhizal structure/total number of microscopic root segments. AMF spores were isolated from 25 g of dry rhizosphere soil by the wet sieving method (the diameters of the upper, middle, and lower sieve holes were 0.71 mm, 0.25 mm, and 0.053 mm, respectively) [22]. Spore morphology was then observed under a 10× stereomicroscope (Leica, Wetzlar, Germany), with the AMF spore number being counted accordingly.

### 2.4. Determination of S. japonica Rhizosphere AMF Diversity

To determine AMF molecular diversity, we first extracted the genomic deoxyribonucleic acid (DNA) from *S. japonica* rhizospheric soil samples according to the instructions provided by the MagaBio soil Kit (BSC48S1E, Bioer, Hangzhou, China). Then, we used the NanoDrop One instrument (Thermo Fisher Scientific, Waltham, MA, USA) to detect DNA purity. Next, the genomic DNA served as a template for polymerase chain reaction (PCR) amplification, using AMV4.5NF (5′-AAGCTCGTAGTTGAATTTCG-3′) and AMDGR (5′-CCCAACTATCCCTATTAATCAT-3′) primers corresponding to the region SSU [23] and TaKaRa Premix Taq Version 2.0 (TaKaRa Biotechnology Co., Dalian, China). The PCR reaction mixture consisted of 25 μL of 2× Novoprotein Taq, 1 μL Primer-F (10 μM), 1 μL Primer-R (10 μM), 50 ng DNA, and ddH_2_O added to a total volume of 50 μL. The PCR reaction conditions were as follows: initial denaturation at 95 °C for 5 min, denaturation at 95 °C for 30 s, annealing at 50 °C for 30 s, and extension at 72 °C for 30 s, for a total of 36 cycles. This was followed by a final extension step at 72 °C for 8 min. The extracted DNA was then stored at 4 °C for preservation. The purity and length of the PCR products were determined by electrophoresis on a 1% agarose gel. Next, equal mass volumes of the PCR products were combined, and we used the E.Z.N.A. Gel Extraction Kit (Omega, Norcross, GA, USA) to recover the PCR amplicons mixture. Then, using Tris-ethylenediaminetetraacetic acid buffer to elute the target DNA fragment according to the procedure of the NEBNext Ultra II DNA Library Prep Kit for Illumina (New England Biolabs, Ipswich, IL, USA) to build the library. The Illumina Nova 6000 platform was used to sequence the amplicon library, and the sequencing service was provided by Guangdong Magigene Biotechnology (Guangzhou, China).

## 3. Data Processing

### 3.1. Sequencing Data Processing

For data processing, we first used Fastp (v0.14.1, https://github.com/OpenGene/fastp (accessed on 25 June 2023)) for sliding-window quality editing of raw data from both terminals (parameter: -W4-M20). Subsequently, the primer sequences located at either end of the sequence were eliminated using Cutadapt (v.1.14, https://github.com/marcelm/cutadapt/ (accessed on 25 June 2023)) for acquiring unblemished paired-end (PE) reads post-quality assurance. Utilizing usearch-fastq_merge pairs (v10.0.240, http://www.drive5.com/usearch/ (accessed on 25 June 2023)) to sift through non-conforming tags, the splice sequence’s initial raw tags were derived based on the overlapping nature of PE reads. This method entailed adjusting certain parameters, such as a minimum 16 bp overlap length and a 5 bp maximum mismatch limit in the splice sequence’s overlap area. Following this, Fastp was employed to execute sliding-window quality clips on the initial label data, which were then processed using Fastp to ensure an efficient clean label. Following this, Usearch (v.10.0240) served to categorize sequences into operational taxonomic units (OTUs). Sequences exhibiting a similarity of 97% or more were categorized under the same OTU. Ultimately, the annotation of OTU species utilized usearch-sintax (v10.0.240), comparing each OTU’s representative sequences with the MaarjAM database (https://www.maarjam.botany.ut.ee (accessed on 25 June 2023)) for comprehensive species annotation details.

### 3.2. Statistical Analysis

AMF community richness, Chao1, Shannon–Wiener, and Simpson indices were computed using Usearch-alpha_div (v 10.0.240, http://www.drive5.com/usearch/ (accessed on 25 June 2023)), offering valuable perspectives on the sample’s community richness and diversity. The variance in soil elements, AMF colonization, spore density, and the alpha diversity index was examined using Excel (v 2016) and Statistical Product and Service Solutions (SPSS, version 26.0). The two groups were tested by two independent-sample t-tests. The three groups were tested by one-way ANOVA and LSD, and the Duncan multiple test (*p* < 0.05). Utilizing R (v 4.2.3) and Excel, boxplots depicting AMF colonization and spore numbers were created, accompanied by a comparative abundance chart of AMF. The metaMDS feature of the vegan package in R (v 4.2.3) was utilized to conduct a non-metric multidimensional scaling (NMDS) analysis, focusing on the OTU level, Bray–Curtis distance algorithm, and Spearman coefficient. The vegan package of R’s anosim function, employing the OTU level and Bray–Curtis distance algorithm, facilitated the examination of molecular variance (AMOVA) and similarity analysis (ANOSIM) to assess the variances in community configurations among groups. Additionally, a Venn figure was created using the venn.diagram feature in R (v 4.2.3) VennDiagram package, serving to determine the count of shared and distinct OTUs in each category. For the Linear discriminant analysis (LDA) Effect Size (LEfSe), the MASS package’s lqa function and the microeco package’s trans-diff function in R (v 4.2.3) were employed to examine notable variances in species across groups, with an LDA score of 2 or higher signifying statistically significant abundance differences. The vegan, ggplot2, ggforce, and ggrepel packages in R (v 4.2.3) were used for canonical correspondence analysis (CCA) to test the relationship between environmental factors and AMF community structure. Furthermore, to assess how environmental elements influence the variance in AMF community distribution, a variance partitioning canonical correspondence analysis (VPA) was conducted using the vegan package’s varpart function in R (v 4.2.3). To evaluate the link between environmental elements and the abundance of AMF species, Spearman’s correlation coefficient was computed, and a correlation heat map was developed to depict these connections. Ultimately, a Mantel test was performed using the qcorrplot feature of the linkET package and the vegan package’s mantel function in R (v 4.2.3), examining the link between the Pearson correlation of environmental elements and AMF genera.

## 4. Results

### 4.1. Rhizosphere Soil Physicochemical Properties of S. japonica

The soil samples of *S. japonica* were generally acidic, with an average pH of 5.61. The average pH of soils developed from dolomite was lower than that of limestone and sandstone. The average contents of AN, AMg, ACa, AK, and SOM in soils developed from limestone are higher than those in dolomite and sandstone. Overall, the average AK, ACa, and AMg contents of *S. japonica* soils were 172 mg kg^−1^, 1454 mg kg^−1^, and 101 mg kg^−1^, respectively. The average content of AK, Aca, and AMg in the soil developed from limestone was 192 mg kg^−1^, 2070 mg kg^−1^, and 151 mg kg^−1^, respectively, which were higher than the average of all samples. The ratio of AK to AMg content in the soil developed by limestone is 1.27, and the ratio of ACa to AMg content is 13.68, which is less than the ratio of all samples. Different physicochemical factors exhibited different trends with altitude. Nonetheless, average pH, AN, ACa, and SOM content increased with altitude. Specifically, at an altitude of 537.40 m, pH and ACa exhibited their relatively lowest values, whereas AK reached its peak value. The physicochemical characteristics of the *S. japonica* soil samples from different soil types and altitudes are presented in the Appendix A.

### 4.2. AMF Colonization and Soil Spore Number of S. japonica

The differences in AMF colonization and spore density of *S. japonica* from different samples are shown in Figure 1 and Figure 2. Figure 1 showed that the average colonization of AMF in the roots of *S. japonica* was 81.11%, and there was no significant difference between different lithology and altitude, but the grouping of soil developed by sandstone at 171.45 m showed a significantly lower colonization of AMF. Figure 2 showed that the average AMF spore density in the rhizosphere soil of *S. japonica* was 1.68 spore per gram soil. The spore density of AMF in soils developed from limestone was significantly different along the altitudinal gradient, and the spore density at 537.40 m altitude was significantly higher than that at other altitudes. The average colonization of AMF in the roots of *S. japonica* in the soil developed from limestone was higher than that in the sandstone and dolomite plots, while the average spore density of AMF in the soil developed from limestone was lower than that in the sandstone and dolomite plots. With the increase in altitude, the AMF colonization of *S. japonica* roots showed a trend of increasing first and then decreasing, increasing between 171.45 and 537.40 m and decreasing between 537.40 and 714.00 m.

### 4.3. Composition and Distribution of the AMF Community in the S. japonica Rhizosphere

The community composition and average relative abundance of the AMF genera at different altitudes and lithological groups are shown in Figure 3. In total, 3,096,236 sequences and 5767 OTUs were identified in *S. japonica* rhizosphere soil. Among these, 270 OTUs were defined at the genus level and divided into 7 genera and 35 species. In the analysis of the top 50 OTUs with average relative abundance, the relative abundance of OTU_1 has certain advantages in each group of samples. At the same time, there were some differences in the distribution of OTUs such as OTU_5 and OTU_2 in each group of samples. The 7 identified genera were as follows: *Paraglomus* (146 OTUs, 2.5%), *Glomus* (98 OTUs, 1.6%), *Claroideoglomus* (9 OTUs, 0.1%), *Diversispora* (9 OTUs, 0.1%), *Archaeospora* (5 OTUs, <0.1%), *Acaulospora* (2 OTUs, <0.1%) and *Gigaspora* (1 OTU, <0.1%). *Paraglomus*, *Glomus*, *Claroideoglomus*, and *Diversispora* were detected across all altitudes, with 22 identified *Glomus* species. Compared with sandstone and dolomite, the average relative abundance of *Glomus* in the soil developed by limestone was the highest, while *Paraglomus* was the opposite.

### 4.4. Rhizosphere Soil AMF Diversity of S. japonica 

We assessed the AMF diversity of *S. japonica* rhizosphere soil using various indices (Table 1). The average richness and Chao1 indices of AMF in the soil samples were determined as 1430 ± 190 and 1431 ± 190, respectively. Further, the average Shannon and Simpson diversity of these AMF communities were 1.76 ± 0.25 and 0.08 ± 007, respectively. There was no significant difference in the Shannon index and Simpson index of the AMF community between sandstone and dolomite groups. The mean values of the Richnesss index and Chao1 index of the AMF community in the rhizosphere soil of *S. japonica* developed from limestone were higher than those of dolomite, while the Shannon index and Simpson index were lower than those of dolomite. NMDS analysis of the AMF community in the rhizosphere soil of *S. japonica* (Figure 4) showed that the distribution of the AMF community was different under different lithologies and altitudes. The stress parameters under different lithology and altitude treatments were 0.1942 and 0.1960, respectively, which were lower than 0.2, indicating that NMDS analysis was reliable. In addition, the *p* values of AMOVA and ANOSIM results under different lithology treatments were 0.00 and 0.048, respectively. The *p*-values of AMOVA and ANOSIM results at different altitudes were 0.00 and 0.001, respectively, indicating that the observed differences between groups were significant.

Venn diagram analysis (Figure 5) revealed 719 common OTUs in the rhizosphere soil of all groups, with the number of distinct OTUs ranging from 100 to 292 per group. Limestone plots exhibited a higher count of distinct OTUs compared to those in dolomite and sandstone. Additionally, the LefSe study (Figure 6) underscored the variances in AMF community structure within the rhizosphere soil of *S. japonica*, varying by lithology and altitude. The species with differing AMF abundances in the rhizospheres across different lithologies included *Glomus Wirsel_OTU6* (sandstone), *G.MO_G18* (dolomite), and *G.MO_G22* (limestone). The species with differing AMF abundances at different altitudes included *G.MO_G18* at an altitude of 100–200 m, *G.Torrecillas12b_Glo_G13* at an altitude of 200–300 m, and *G.Glo3b*, *G.MO_G22*, and *G.MO_G8* at an altitude of 700–800 m. Overall, we primarily identified 6 *Glomus* species with differential AMF abundances in rhizosphere soils with different lithologies and altitudes.

### 4.5. Relationship between AMF and Environmental Factors 

CCA results (Figure 7) showed that environmental factors such as altitude, pH, and lithology were closely related to the AMF community, which were important factors affecting the distribution of AMF in the rhizosphere soil of *S. japonica*. CCA1 and CCA2 explained 17.0% and 15.4% of the total variables, respectively. Additionally, VPA (Figure 8) identified an interpretation rate of 3% regarding the contribution of pH and AP to the variation. Moreover, altitude and lithology were determined to exhibit interpretation rates of 2% and 1%, respectively. Notably, the shared influence of pH, AP, and altitude was determined at 1% and an unidentified portion was determined at 92%. Overall, lithology, altitude, pH, and AP all play an important role in the formation of *S. japonica* rhizosphere AMF, with pH and AP emerging as leading factors.

Next, correlation analysis was conducted between species abundance of AMF present in the rhizosphere of *S. japonica* and the environmental factors (Figure 9). Overall, the sensitivity of each AMF to different soil physicochemical properties differed. The contents of SOM, AN, and ACa were positively correlated with the abundance of OTU_1, OTU_4, and OTU_6. The contents of AP and AK were negatively correlated with the abundance of OTU_3, OTU_95, and OTU_11. Altitude was positively correlated with the abundance of *Acaulospora*, and negatively correlated with the abundance of *Gigaspora* and *Paraglomus*. Over-enrichment of SOM and AN may reduce the abundance of certain species, including *Gigaspora* and *Diversispora*. The corresponding cluster tree diagram showed that SOM and AN were closely related. There was a positive correlation between pH and species abundance including *Glomus* and *Acaulospora*.

### 4.6. The Association between Environmental Factors and AMF Characteristics 

The association between AMF community changes and environmental factors is shown in Figure 10. The effects of physicochemical factors and pH on AMF were determined to be complex and variable. Altitude, the content of SOM, Aca, and AN were positively correlated with AMF colonization, the community Shannon index, and the Chao1 index. There was a positive correlation among SOM, Aca, and AN (*p* < 0.01). Alternatively, rhizosphere soil pH was negatively correlated with AMF colonization (*p* > 0.05) and spore density (*p* < 0.01), but positively correlated with AMF Shannon and Chao1 indices (*p* > 0.05). The contents of AP, AK, and AMg were negatively correlated with the Shannon and Chao1 indices of the AMF community but positively correlated with AMF colonization (*p* > 0.05).

## 5. Discussion

### 5.1. Characteristics of the AMF Community in the Rhizosphere of S. japonica 

This study revealed the AMF germplasm resources in the rhizosphere of *S. japonica*, and 22 species were identified by *Glomus*. *Glomus* has also been identified as a predominant genus in the rhizosphere of other medicinal plants, including *Rosa laevigata* Michx [24] and *Siraitia grosvenorii* [25]. These results may, ultimately, be attributed to the adaptive characteristics of *Glomus*. *Glomus*, belonging to the Glomeraceae family, can effectively and rapidly establish connections with infected roots and free mycelial fragments of plants [26,27]. This not only leads to enhanced root colonization [28] but also contributes to a larger proportion of mycelium abundance compared to that of other AMF families [26,29]. Additionally, Glomeraceae have a short sporulation time, low soil biomass allocation, and high root biomass allocation, and are less susceptible to soil disturbances [26]. Even after interference, Glomeraceae can quickly reconstruct functional hyphae; this has been attributed to the high mycelium and tissue turnover rates [26], which confer enhanced host protection. Compared to other AMF, *Glomus* spores exhibit faster growth, with *Glomus* spp. investing considerably in spore formation [26,27]. Additionally, these spores are more prevalent in soils with high disturbance and low nutrient content [30]. It has been postulated that frequently disturbed farmland soils are primarily dominated by *Glomus*. Chagnon et al. suggested that this may serve as a strategy to avoid interference [26]. Moreover, *Glomus* is generally considered to be the dominant genus in karst rocky desertification regions, which may be attributed to its short life cycle, high growth rate [13,26], strong ecological adaptability [7], and potential for ecological restoration [16,31]. Considering both the findings from prior studies and those of the current study, it can be hypothesized that *Glomus* spp. may be particularly suitable for farmland soils within karst rocky desertification habitats, potentially establishing themselves as the dominant genus in such environments. Therefore, future research should combine metagenomic and genetic strategies to analyze the microbial function of *Glomus* spp., exploring their ecological advantages, response mechanisms, and net benefits in different habitats.

### 5.2. Correlation between Altitude and AMF Community of S. japonica

This study focused on the impact of altitude (100–800 m) on AMF colonization and rhizosphere community dynamics of *S. japonica*. Overall, an increase in altitude increased AMF colonization, community richness, and diversity of *S. japonica*–associated soil. These trends align with those observed when evaluating AMF community diversity and total colonization of *Artemisia ordosica* rhizospheres [32], and AMF diversity in the Helan Mountain region [11]. The positive impact of altitude on the AMF community may be attributed to two key factors. First, this altitude-associated increase in the AMF community richness may be linked to the adaptability of *Acaulospora* spp. For example, Acaulosporaceae is the predominant AMF family identified in the high-altitude Tibetan Plateau, with its relative abundance increasing with altitude [33]. Moreover, *Acaulospora* spp. are more abundant in mountainous areas with high-altitude temperate climates [13,33]. Notably, *Acaulospora* spp. may exhibit enhanced resistance to external abiotic stressors by minimizing their biomass consumption [13,26]. In high altitudes and extreme environments, these species also exhibit greater adaptability and stronger resistance than other AMF genera [13,26]. Therefore, the resistance of *S. japonica* to extreme high-altitude environments may be associated with the homeostasis of *Acaulospora* spp. in the rhizosphere. Second, plants require AMF to facilitate the alleviation of environmental stress. This is particularly relevant in high-altitude areas, with environmental stressors such as low temperature, low humidity, and high UV radiation. These factors reduce the effective accumulated temperature required for plant growth and shorten the growing season, resulting in limited plant growth at high altitudes [12,34]. Host plants seek cooperation from AMF to cope with this increased environmental stress, increasing the opportunity for AMF to contact the host. The increase in AMF diversity at high altitudes may result from shared environmental selection pressures, with host plants adapting to the needs of high-altitude environmental conditions via changes in the AMF community composition [12,33]. These ecological factors also regulate species competition and resource allocation in AMF communities [12,33], indirectly improving AMF competitiveness [35].

Spore density provides insights into the potential reproduction and dispersal abilities of AMF. Spore numbers in *S. japonica* rhizosphere soil fluctuated with increasing altitude; this could be attributed to several factors. First, the nutrient conditions in the transition zone may affect the survival of AMF spores. AMF spore density was the highest at 537.4 m above sea level and then exhibited a decreasing trend. This fluctuation may be because the soil conditions at this altitude are in the transition zone, where the reproductive capacity and species diffusion distribution of AMF are significantly affected by altitude and soil conditions. Second, elevation may be unfavorable for spore survival in some species. Certain species, including Gigasporaceae and Paraglomeraceae spp., gradually diminish with increasing altitude, which is accompanied by various environmental stresses including low temperature, low humidity, and intense UV radiation [12]. The presence of Paraglomeraceae, for example, is more common in low-altitude areas but becomes scarcer at higher altitudes [12]. This aligns with the negative relationship between altitude and Gigasporaceae and Paraglomeraceae abundance observed in the present study. An increase in altitude may reduce the available AMF niche [13]; this can result in a reduced abundance of Gigasporaceae, an AMF family that has been determined to provide greater benefits for the host and effectively utilize soil resources [13,26]. Ultimately, the absence of these functional species may contribute to the corresponding fluctuations in spore numbers. Finally, AMF produce propagules under high-altitude stress conditions, possibly due to the positive response of spores to environmental stress and the various survival strategies employed by AMF [10]. The broad ecological adaptability of *Glomus* spp. influences their ability to survive under stressful environments, such as tropical rainforests, deserts, and karst rocky desertification regions [25]. Therefore, exploring the relationship between altitude and the reproduction and dispersal ability of functional species will provide a basis for the preservation and application of mycorrhizal fungal germplasm resources.

### 5.3. Correlation between Lithology and AMF Community of S. japonica

We investigated AMF communities in the *S. japonica* rhizosphere across three different lithologies: sandstone, dolomite, and limestone. Overall, the unique habitat of karst limestone soil was determined to be more conducive to *S. japonica* roots AMF colonization. This finding can be attributed to two possible factors. First, the rock properties of limestone determine that its physical and chemical weathering rate exceeds that of dolomite [36,37,38], which promotes the increase in soil organic carbon, total nitrogen, and other nutrients [38], which is conducive to plant growth. Simultaneously, the well-developed plant roots provide favorable host conditions for AMF colonization. Specifically, in this study, the contents of AN, ACa, SOM, and other nutrients in the soil developed from limestone were higher than those in dolomite and sandstone, which may lead to the high AMF colonization of *S. japonica* roots in the soil developed from limestone. Second, the exposure rate of the limestone bedrock is higher than that of dolomite, indicating that limestone soil is more discontinuous than dolomite [36]. Owing to the irregular distribution of limestone soil and strong karstification, diversified microhabitats, such as grooves, stone ditches, and stone cracks, develop more frequently on the rock surface [38]. These microhabitats affect the erosion of soil nutrients by surface runoff [39,40,41]. Such soil conditions may be more suitable for the development process of AMF communities, offering diverse habitats for AMF, thereby increasing AMF abundance [38]. These findings align with those of past research, which demonstrated that the AMF richness of limestone soil was higher than that of dolomite in secondary forests, shrubs, farmlands, and grass ecosystems [8]. In summary, lithology plays a crucial role in regulating soil surface various biological processes, including the flow of soil elements, nutrient transfer, and resource acquisition [6,7,38]; therefore, lithology is an important factor in regulating plant growth and shaping the microbial community.

### 5.4. The Relationship between AMF Community and Soil Characteristics

Abiotic stresses, such as high soil acidity, have often been identified as important AMF community driving factors [26]. Nonetheless, under the harsh pH conditions of the *S. japonica* rhizosphere soil, AMF richness and diversity exhibited a positive response; this could be attributed to the niche and strain tolerance of specific taxonomic groups, including *Glomus*, *Claroideoglomus*, and Acaulosporaceae spp. Specifically, low soil pH and harsh climatic conditions have been determined as favorable for the survival and sporulation of Acaulosporaceae spp. [26,31,42]. Moreover, *Glomus* is a well-known genus with a high growth rate and strong adaptability [25]. Further, *Claroideoglomus,* with its wide niche distribution in various areas, such as forestland, grassland, and shrubland, can gain an advantage in the early and mid-successional stages [43]. These advantages may be attributed to the positive feedback loop between AMF and acidic soil stress. This understanding aligns with the results of a study that evaluated the pH of *Siraitia grosvenorii* in Guangxi; in this study, an increase in AMF richness and diversity was observed with increasing pH [25]. Therefore, exploring the sensitivity of mycorrhizal fungi to soil pH and evaluating how soil pH drives microbial niche differentiation in rocky desertification areas could offer innovative approaches to harness biodiversity for the mitigation of rocky desertification and facilitation of ecological restoration.

AN and SOM both increased AMF colonization in *S. japonica* roots and enhanced AMF richness and diversity. This may be because nitrogen enrichment can improve the carbon sequestration potential of plants, allowing them to provide more photosynthetic products to AMF [26]. This, in turn, synergistically promotes plant growth and AMF colonization. However, in a study on Taibai Mountain in the Qinling Mountains, C and N levels were determined as negatively correlated with the AMF Shannon index; this could be attributed to the decrease or loss of some fungal species caused by changes in soil nutrients, accompanied by the increase in dominance of other species [14]. Soil AN and SOM were negatively correlated with the abundance of Gigasporaceae and *Diversispora*. *Diversispora* spp. have a large quantity of external mycelia, which are more suitable for survival in low-nutrient soils such as grasslands and shrubs [44,45]. Gigasporaceae primarily reproduce via spore regeneration, which is the slowest known AMF colonizing method; however, these Gigasporaceae spores decrease with nitrogen enrichment [28]. Therefore, AN enrichment in the rhizosphere of *S. japonica* can facilitate SOM accumulation and synergistically promote plant growth and mycorrhizal fungal colonization. Nonetheless, the sensitivity of different AMF species to nitrogen content varies [16]. Excessive enrichment and accumulation of C and N may reduce the abundance of certain species, and can interfere with AMF adaptability to the soil environment [14].

Soil phosphorus and potassium are also important factors in predicting AMF diversity and abundance [14]. Low AP levels have been linked to an increase in AMF community diversity, while a small amount of potassium is also considered a prerequisite for effective AMF colonization in certain plants [46]. Some researchers have hypothesized that low AP may promote AMF abundance [16] and that AMF can promote the absorption of potassium by olives [47]. However, plants may struggle to establish an effective symbiotic relationship with AMF under enrichment or deficiency of soil AK [46]. Moreover, with an increase in the soil phosphorus utilization rate, AMF diversity has been shown to exhibit a downward trend [14]. Soil AP and AK inhibited AMF diversity in the *S. japonica* rhizosphere to varying degrees. This could be attributed to the high content of AP and AK, which limited resource exchange between AMF and the host; therefore, the nutrients obtained by AMF could not be adequately supplied to the host, which had an adverse effect on AMF stability and reproduction in the *S. japonica* rhizosphere. Prior studies have indicated that spore density may be linked to the diffusion rate of potassium and that the slow diffusion of K ions is conducive to spore germination; this suggests that soil K may be positively correlated with spore density [48]. However, this contrasts with the negative correlation between AK and spore numbers observed in the present study. Therefore, the weathering of karst rocks was postulated to affect the diffusion and loss rates of soil phosphorus and potassium, potentially limiting their effective uptake by plants, thereby restricting the colonization and community development of mycorrhizal fungi. However, further experimental evidence is required to confirm this hypothesis.

Overall, calcium is the main factor that determines the stability of SOM, C, and N levels [49]. The difference in the chemical composition of limestone (CaCO_3_) and dolomite (CaMg(CO_3_)_2_) indicates that the calcium content of limestone is higher than that of dolomite [37]. Ultimately, this contributes to the accumulation of SOM and nitrogen within limestone-derived soils [49]. The contents of SOM, AN, and ACa in the rhizosphere soil of *S. japonica* were significantly positively correlated (*p* < 0.01); this aligned with the results of a similar study conducted in the karst region of Guizhou, China [36]. ACa in the rhizosphere of *S. japonica* played a positive role in AMF colonization and community richness. It was hypothesized that the high calcium content in the limestone soil provides a suitable foundation for soil SOM accumulation and nitrogen saturation. This, in turn, facilitates the uptake of sufficient carbon and nitrogen by plants, thereby promoting plant growth and increasing the supply of photosynthetic products for mycorrhizal fungi. Additionally, the method by which calcium forms complexes with SOM in limestone areas provides a stable soil environment for mycorrhizal fungi colonization and community development, improving the richness and diversity of AMF. However, despite higher carbon and nitrogen content in limestone areas, the unique structure of thin and discontinuous soil layers can result in rapid nutrient loss [50] Some studies have suggested that soil calcium nutrients decrease as soil carbon is consumed [51]. Ultimately, future research should explore the regulatory mechanisms of nutrient cycling and degradation in karst limestone soil with the aim of using the ecological benefits of mycorrhizal fungi to reduce the requirement for chemical fertilizer use in agriculture.

Soil-exchangeable magnesium, the primary form of magnesium available for crops, is an important indicator of soil fertility [31]. Magnesium and AMF richness, diversity and spore density showed a negative correlation; this may be attributed to the magnesium content of *S. japonica* soil being lower than that of calcareous soil in arid or semi-arid areas (20 g kg^−1^ average) [31]. Previously, it has been established that the AK/AMg ratio in limestone is >0.6, and the ACa/AMg ratio is >10 [52,53]. Additionally, a potential lack of magnesium in the soil has been established to affect plant growth and mycorrhizal fungal community stability [52,53]. Although the AK/AMg and ACa/AMg results demonstrated that the rhizosphere soil of *S. japonica* contained limited magnesium, some research has demonstrated that AMF can promote the absorption of soil magnesium by the roots. Moreover, this effect was determined to be more significant under moderate magnesium deficiency [53]. This research is similar to the results of this study, which indicated that AMF colonization is positively correlated with the AMg content, which may be a positive response of AMF to magnesium-limited adversity. These results corroborate the hypothesis that plants benefit more from AMF symbiosis under low-fertility conditions [10,54,55]. Overall, future studies should explore the mechanism by which soil magnesium supply and migration antagonistically or synergistically regulate mycorrhizal fungal community dynamics; further, other soil physicochemical factors should be explored at different plant growth stages to research the effect of mycorrhizal fungi.

## 6. Conclusions

This study explored *S. japonica* rhizosphere AMF diversity across various lithologies and altitudes in the karst regions of Guangxi, China. The corresponding results demonstrated that the AMF community in *S. japonica* rhizosphere consisted of 7 genera and 35 species, including *Glomus*, *Paraglomus*, and *Acaulospora*, and germplasm resources were abundant. Lithology, altitude, pH, and AP were established as important factors driving AMF colonization and community dynamics within *S. japonica*–associated soil. Moreover, we established that there may be a coupling effect between AN, SOM, and ACa in limestone-derived soil; this relationship was important for predicting soil carbon and nitrogen accumulation, changes in community diversity, and nutrient cycling. The mechanisms responsible for the interrelationship between plants and environmental factors, alongside the function of AMF strains and soil factors, require further experimental evidence. Future research should explore the interaction mechanism between indigenous and dominant strains and soil factors, propagate strains for inoculation experiments, explore the potential survival mechanism of AMF in rocky desertification areas, and promote the ecological cultivation of medicinal plants by using the species association of medicinal plant fungal chaperones to reduce the dependence on chemical fertilizers in the future. Our study highlights the potential for leveraging the biodiversity and species associations of karst mycorrhizal fungi to promote the effective ecological restoration of mycorrhizal *S. japonica* seedlings. Moreover, this study provides novel strategies for mitigating rocky desertification.

## Figures and Tables

**Figure 1 jof-10-00340-f001:**
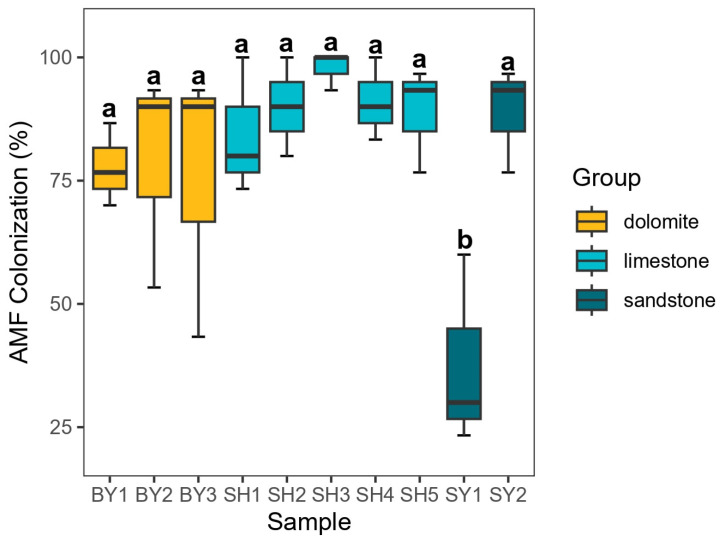
AMF colonization of *Sophora japonica* roots. Note: SY represents sandstone; SY1 and SY2 represent altitudes of 171.45 and 672.25 m, respectively. BY represents dolomite; BY1, BY2, and BY3 represent altitudes of 171.45 and 230.40 m, respectively. SH represents limestone; SH1, SH2, SH3, SH4, and SH5 represent altitudes of 319.10, 438.50, 537.40, 672.25, and 714.00 m, respectively. The difference in AMF colonization in each sampling group of *S. japonica* was represented by different lowercase letters (*p* < 0.05).

**Figure 2 jof-10-00340-f002:**
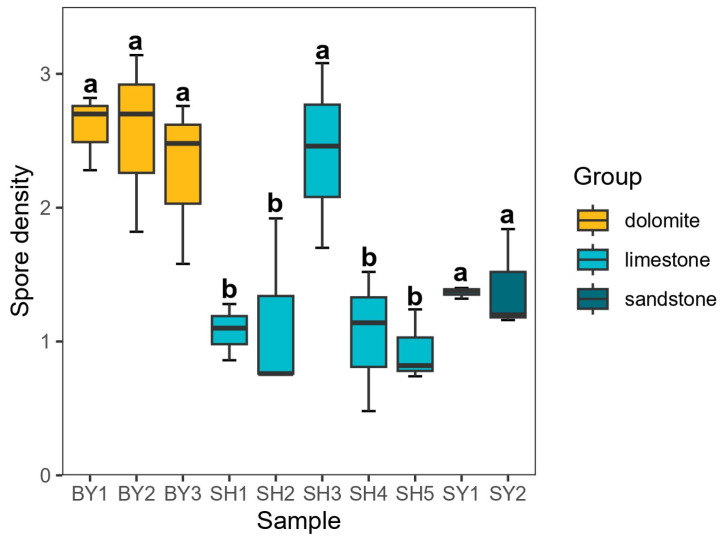
Density of AMF spores in rhizosphere soil of *Sophora japonica*. Note: SY represents sandstone; SY1 and SY2 represent altitudes of 171.45 and 672.25 m, respectively. BY represents dolomite; BY1, BY2, and BY3 represent altitudes of 171.45 and 230.40 m, respectively. SH represents limestone; SH1, SH2, SH3, SH4, and SH5 represent altitudes of 319.10, 438.50, 537.40, 672.25, and 714.00 m, respectively. The difference in AMF spore density in each sampling group of *S. japonica* was represented by different lowercase letters (*p* < 0.05).

**Figure 3 jof-10-00340-f003:**
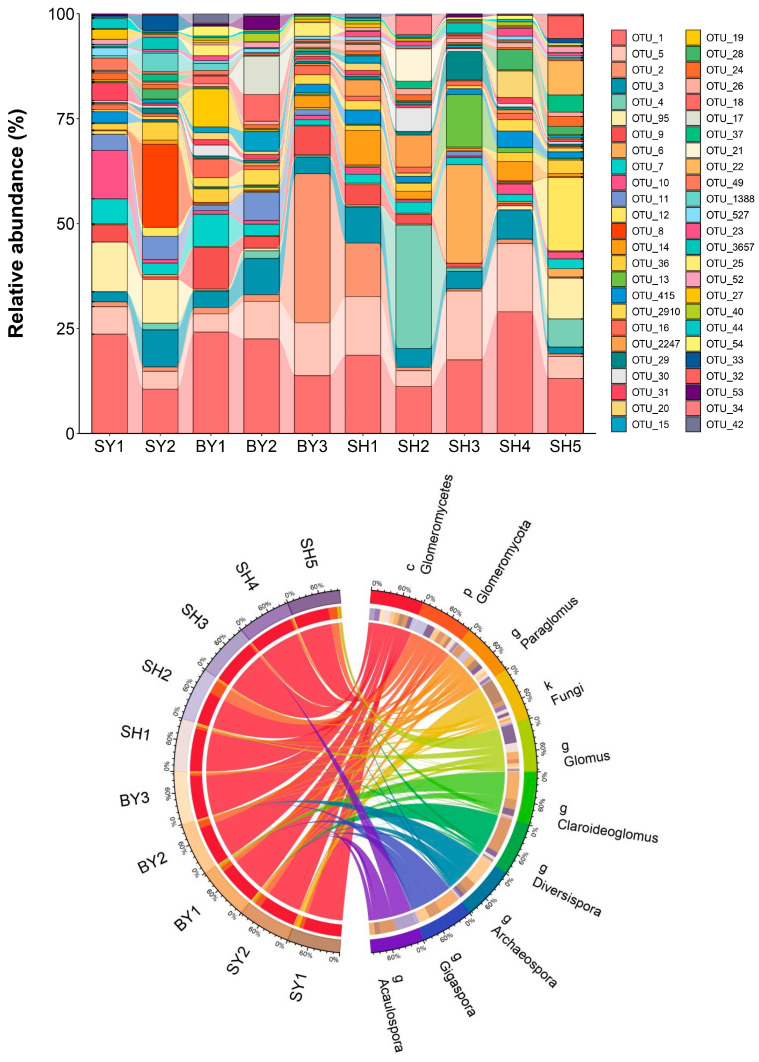
The average relative abundance of top 50 OTUs and AMF genera in *Sophora japonica* rhizosphere soil. Note: SY represents sandstone; SY1 and SY2 represent altitudes of 171.45 and 672.25 m, respectively. BY represents dolomite; BY1, BY2, and BY3 represent altitudes of 171.45 and 230.40 m, respectively. SH represents limestone; SH1, SH2, SH3, SH4, and SH5 represent altitudes of 319.10, 438.50, 537.40, 672.25, and 714.00 m, respectively. The left side of the circle indicates sample names, while the right side indicates the AMF. A connection between a group and a specific classification indicates the presence of AMF in the sample. A thicker connection indicates a higher species abundance. k: kingdom; p: phylum; c: class; g: genus.

**Figure 4 jof-10-00340-f004:**
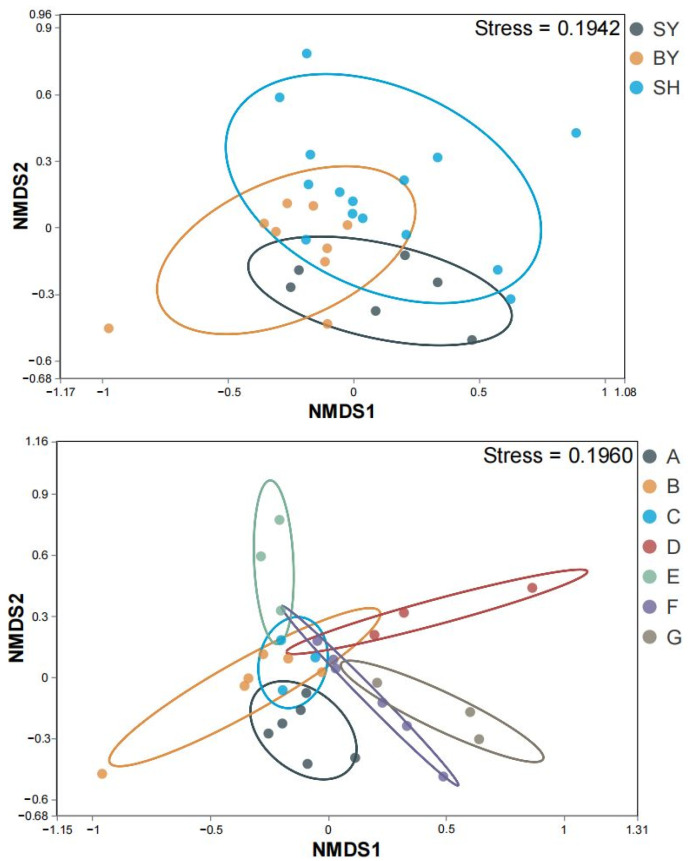
Rhizosphere soil AMF NMDS analysis of *S. japonica.* Note: SY represents sandstone; BY represents dolomite; SH represents limestone. A, B, C, D, E, F, and G represent altitudes of 171.45, 230.40, 319.10, 438.50, 537.40, 672.25, and 714.00 m, respectively. NMDS: non-metric multidimensional scaling. Different colors represent different groups.

**Figure 5 jof-10-00340-f005:**
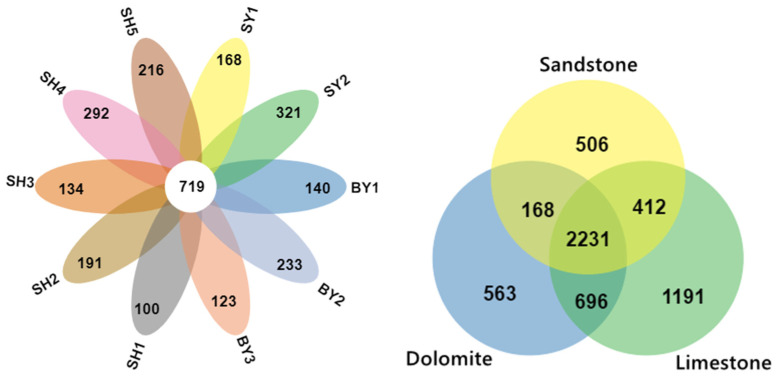
AMF Venn diagram of the OTUs within the rhizosphere of *S. japonica* cultivated in different soil types and at different altitudes. Note: SY represents sandstone; SY1 and SY2 represent altitudes of 171.45 and 672.25 m, respectively. BY represents dolomite; BY1, BY2, and BY3 represent altitudes of 171.45 and 230.40 m, respectively. SH represents limestone; SH1, SH2, SH3, SH4, and SH5 represent altitudes of 319.10, 438.50, 537.40, 672.25, and 714.00 m, respectively. Different colors represent different communities. Overlapping areas represent the number of OTUs shared by the group, while non-overlapping regions represent the number of different OTUs within the group.

**Figure 6 jof-10-00340-f006:**
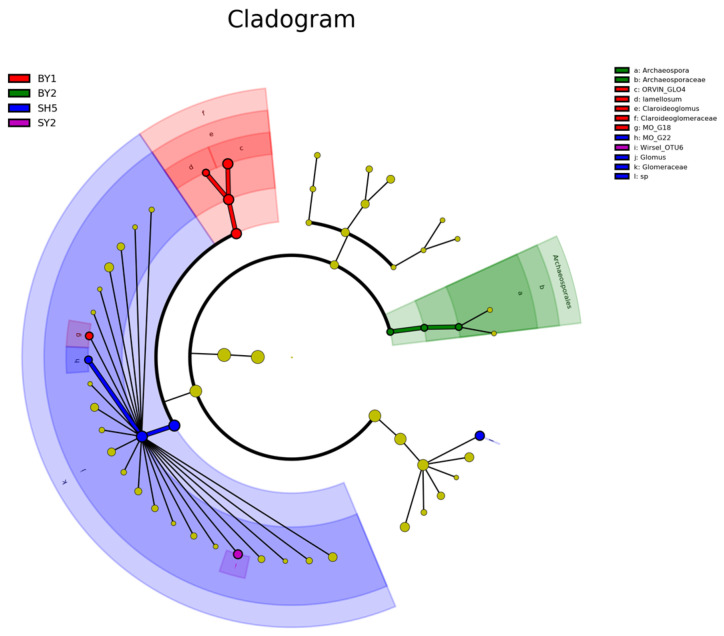
LEfSe analysis of AMF in *S. japonica* rhizosphere soil cultivated in different soil types and at different elevations. Note: SY represents sandstone; SY2 represents an altitude of 672.25 m. BY represents dolomite; BY1 and BY2 represent altitudes of 171.45 and 230.40, respectively. SH represents limestone; SH5 represents an altitude of 700–800 m. A, B, C, E, and G represent altitudes of 171.45, 230.40, 319.10, 537.40, and 714.00 m, respectively. LEfSe: Linear discriminant analysis Effect Size.

**Figure 7 jof-10-00340-f007:**
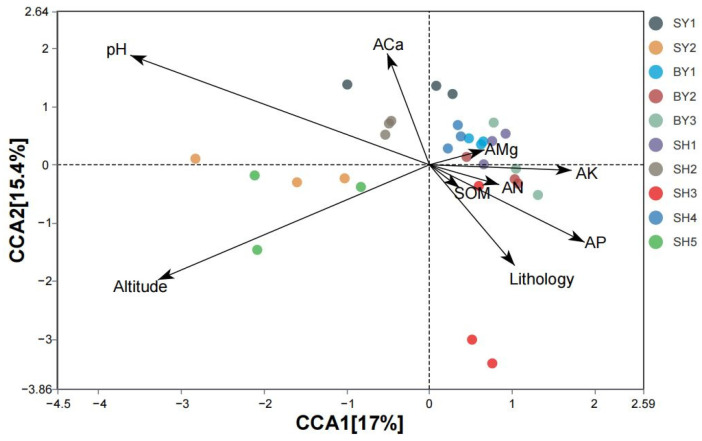
CCA of environmental factors and *S. japonica* rhizosphere AMF. Note: different arrows represent distinct environmental factors and different points represent individual samples. The longer the arrow, the greater the effect on the sample. The sharp angle between the arrows is positively correlated with environmental factors, and the obtuse angle is negatively correlated. The sample is projected onto an environmental factor or its extension line; the relative position of the projection point represents the relative size of the environmental factor value in the sample. SY represents sandstone; SY1 and SY2 represent altitudes of 171.45 and 672.25 m, respectively. BY represents dolomite; BY1, BY2, and BY3 represent altitudes of 171.45 and 230.40 m, respectively. SH represents limestone; SH1, SH2, SH3, SH4, and SH5 represent altitudes of 319.10, 438.50, 537.40, 672.25, and 714.00 m, respectively. AP: available phosphorus. AN: available nitrogen. AMg: available magnesium. ACa: available calcium. AK: available potassium. SOM: soil organic matter.

**Figure 8 jof-10-00340-f008:**
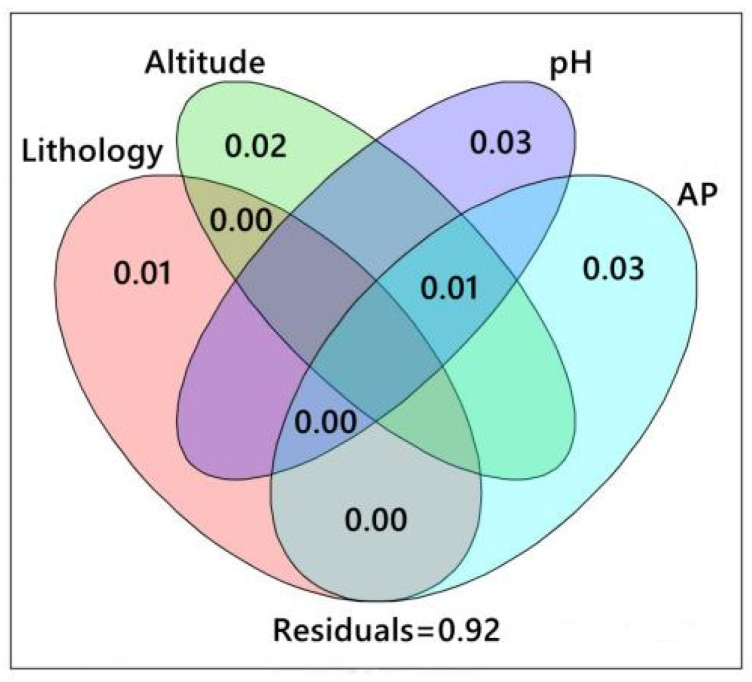
Impact of environmental variables on *S. japonica* AMF community. Note: each oval represents a distinct environmental factor. The percentage overlap represents the degree of interpretation following the interaction of two (or more) environmental factors, excluding other effects. The percentage of non-overlapping parts represents the degree of interpretation attributed to a specific environmental factor for the difference in the distribution of AMF community species after the exclusion of other factors. Outside the circle represents the proportion of all unexplained environmental factors that contribute to AMF community composition.

**Figure 9 jof-10-00340-f009:**
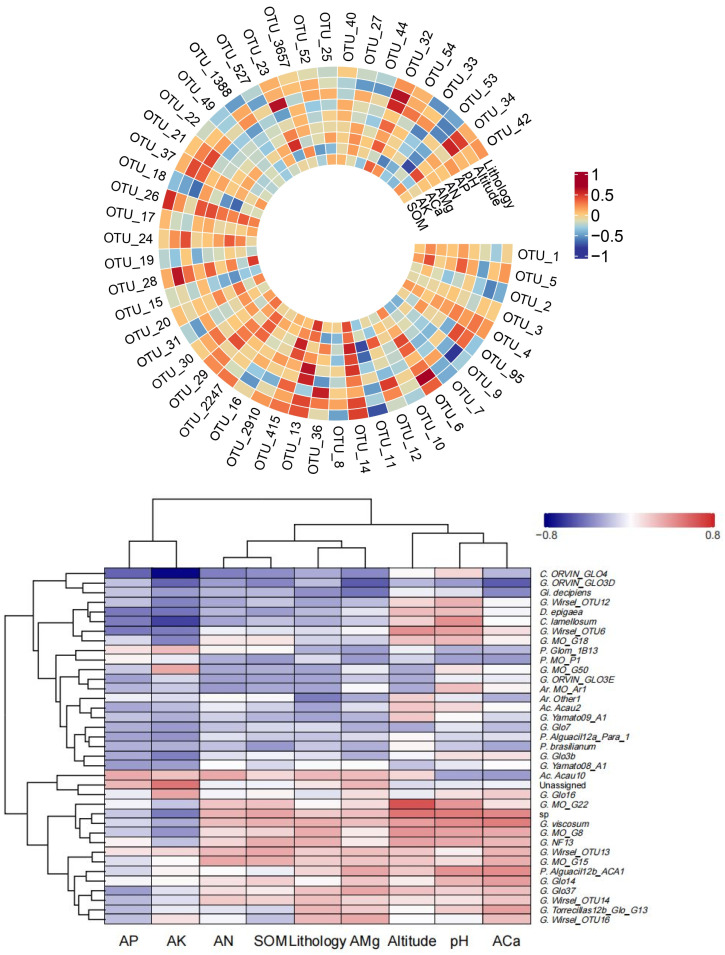
Spearman correlation heat map of the average relative abundance of top 50 OTUs and AMF species level and environmental factors. Note: the horizontal rows represent the AMF and the vertical columns represent the phenotypic information of the sample. Red and blue represent positive and negative correlations, respectively. The depth of the color represents the level of correlation. AP: available phosphorus. AN: available nitrogen. AMg: available magnesium. ACa: available calcium. AK: available potassium. SOM: soil organic matter. *P.*: *Paraglomus. G.*: *Glomus. C.*: *Claroideoglomus. Gi.*: *Gigaspora. D.*: *Diversispora. Ac.*: *Acaulospora. Ar.*: *Archaeospora*. sp.: no species-level information. Unassigned: unable to classify.

**Figure 10 jof-10-00340-f010:**
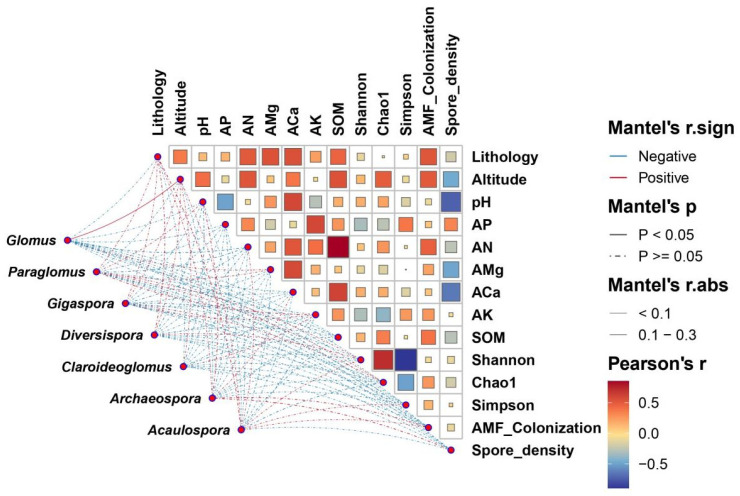
Pearson correlation relationship between AMF properties and environmental factors. Note: the line color of the network diagram represents the correlation between each AMF genus and ecological factor; red and blue represent positive and negative correlations, respectively. The line type of the network diagram represents the *p*-value range; the solid represents *p* < 0.05 and the dot-dash represents *p* ≥ 0.05. The line thickness represents the absolute value of the correlation coefficient. Correlation Heat map color represents the correlation of ecological factors. AP: available phosphorus. AN: available nitrogen. AMg: available magnesium. ACa: available calcium. AK: available potassium. SOM: soil organic matter.

**Table 1 jof-10-00340-t001:** Index of AMF community alpha diversity in rhizosphere soil of *Sophora japonica.*

Sample ID	Richness Index	Chao1 Index	Shannon Index	Simpson Index
SY1	1204 ± 93 b	1205 ± 93 b	1.69 ± 0.06 a	0.07 ± 0.01 a
SY2	1676 ± 204 a	1676 ± 203 a	1.92 ± 0.28 a	0.06 ± 0.06 a
BY1	1442 ± 32 ab	1443 ± 32 ab	1.86 ± 0.06 a	0.06 ± 0.02 a
BY2	1558 ± 113 a	1559 ± 113 a	1.90 ± 0.04 a	0.05 ± 0.01 a
BY3	1268 ± 190 b	1270 ± 189 b	1.56 ± 0.58 a	0.17 ± 0.22 a
SH1	1383 ± 128 bc	1385 ± 128 bc	1.79 ± 0.15 ab	0.06 ± 0.02 a
SH2	1319 ± 105 c	1320 ± 105 c	1.65 ± 0.25 ab	0.09 ± 0.06 a
SH3	1271 ± 89 c	1273 ± 89 c	1.49 ± 0.09 b	0.12 ± 0.01 a
SH4	1627 ± 117 a	1627 ± 117 a	1.92 ± 0.09 a	0.05 ± 0.01 a
SH5	1555 ± 41 ab	1556 ± 41 ab	1.85 ± 0.19 a	0.06 ± 0.04 a

Note: the value represents the mean ± standard deviation. The difference in AMF community alpha diversity in each sampling group of *Sophora japonica* was represented by different lowercase letters (*p* < 0.05).

## Data Availability

The raw reads have been stored in the NCBI Sequence Read Archive (SRA) database (PRJNA1051012). Temporary Submission ID: SUB14032559.

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
