# Peer review of "Arbuscular Mycorrhizal Fungi Diversity in Sophora japonica Rhizosphere at Different Altitudes and Lithologies"

_jof, 2024, doi:10.3390/jof10050340_

Round 1
Reviewer 1 Report
The manuscript presents the results of a survey of arbuscular mycorrhizal fungus (AMF) diversity in the rhizosphere of Sophora japonica in sites in differing altitudes and soils originating from limestone, dolomite, and sandstone. The topic is of interest to AMF ecology and diversity studies. The methods are well described and coherent with this exploratory survey's goals. The results are well presented, but the tables and figures need improvement. The Discussion is, as a whole, well-founded, but some parts are speculative. See observations below and in the annotated text.
The Conclusion section mixes a summary of results with conclusions, i.e., the main phenomena and the underlying factors that determine them. That is clear in lines 674-678, but the authors must condense this section to underline their main findings and present some hypotheses. It is not a good idea to present the limitations of the work so bluntly; the readers know that, and that should not be underlined. It is preferable to present new hypotheses and lines of investigation.
Conclusion:
This section mixes a summary of results with conclusions, i.e., the main phenomena and the underlying factors that determine them. That is clear in lines 674-678, but the authors must condense this section to underline their main findings and present some hypotheses. It is not a good idea to present the limitations of the work so bluntly; the readers know that, and that should not be underlined. It is preferable to present new hypotheses and lines of investigation.
Abstract
Lines 23-25: This sentence is not necessary. The results stand for themselves.
Introduction:
Line 29: Replace kind with group.
Line
Review and condense the final paragraph, ending with your aims.
Materials and Methods
Lines 123-141: This can be shortened. Cite the methods (e.g., Walkely-Black for SOM), adding some general information and any modifications from the standard procedures.
Lines 149-165: Same as above, condense the information.
The site identification should be easily relatable to the lithology. The reader can easily remember that areas identified with SY are on oils originating from sandstone, but that is not the case for the areas with soils from dolomite (BY) or limestone (SH). The goal is to reach a broad audience in Biology and Agricultural Sciences, so it is more appropriate to use D and L, respectively.
Results
In all tables and Figures, the order of treatments varies. Please use the same order for the sites in all Tables and Figures.
Line 244 and others: Consider significant digits in your quantitative data. It is hard to believe that the measurements were that accurate; even if that were the case, it would be distracting and below the significant differences between treatments.
Lines 264-266: That section is redundant and should be suppressed.
Figure 3: Identify altitudes with values or numbers such as 1-2 to 100-200 m, 2-3 to 200-300m, and so forth.
Line 324: The term “held reliability” sounds odd. Does it mean “is reliable”?
Line 366: Separate “is”
Discussion:
Section 5.1 discusses the occurrence of Glomus, but Paraglomus had many OTUs, which should be addressed.
Lines 445-446: This sentence is redundant.
Line 463: Replace suggesting with suggested.
Lines 545=556: Since there were no significant differences in the diversity index, this section is speculative and should be removed, as it is not based on the data the authors had.
Lines 561-568: This section is redundant.
Line 633: The authors state that the rhizosphere significantly correlates with several variables. It must be specified which rhizosphere traits correlate with those variables.

Author Response
Dear reviewer, I express sincere thanks to you on behalf of all the authors. We are very fortunate to meet professional, responsible, patient and careful experts like you. This is very important for the improvement of our manuscript quality. It also broadens our academic thinking and enhances our ability to express. We have seriously considered the reviewer 's suggestions and carefully read the annotations in the document. We are very grateful for your detailed guidance on the manuscript. We have revised the article abstract, introduction, method, picture, table, result statement, etc. The English expression, unit symbols, method reference and details, picture beauty and clarity of the manuscript have been greatly improved. Thank you again for the professional guidance, thank you for your sincere help.
Comments 1: Conclusion: This section mixes a summary of results with conclusions, i.e., the main phenomena and the underlying factors that determine them. That is clear in lines 674-678, but the authors must condense this section to underline their main findings and present some hypotheses. It is not a good idea to present the limitations of the work so bluntly; the readers know that, and that should not be underlined. It is preferable to present new hypotheses and lines of investigation.
Response 1: Dear reviewer, thank you for your advice. We have seriously considered the expression of this part, and revised the research limitations and future research directions of the conclusion part according to your suggestions.
Comments 2: Abstract: Lines 23-25: This sentence is not necessary. The results stand for themselves.
Response 2: Thank you for your suggestions, we have modified the abstract, deleted the unnecessary information, and added more detailed expressions.
Comments 3: Introduction: Line 29: Replace kind with group.
Response 3: Thank you for your detailed review, we have revised the word
Comments 4: Review and condense the final paragraph, ending with your aims.
Response 4: We have modified the last paragraph of the introduction to make it more concise.
Comments 5: Materials and Methods: Lines 123-141: This can be shortened. Cite the methods (e.g., Walkely-Black for SOM), adding some general information and any modifications from the standard procedures.
Response 5: Thank you for your advice. We have concisely described the determination methods of soil physicochemical properties and added literature for readers ' reference.
Comments 6: Lines 149-165: Same as above, condense the information.
Response 6: We have modified the detection methods of mycorrhizal colonization and spore density to avoid occupying too much space in the article.
Comments 7: The site identification should be easily relatable to the lithology. The reader can easily remember that areas identified with SY are on oils originating from sandstone, but that is not the case for the areas with soils from dolomite (BY) or limestone (SH). The goal is to reach a broad audience in Biology and Agricultural Sciences, so it is more appropriate to use D and L, respectively.
Response 7: Thank you for your advice. We have carefully considered your proposal, and the sample ID in the manuscript can be changed. However, considering that the information number we uploaded to the NCBI database will be inconsistent with the changed number, it may be difficult for readers to correspond one by one. In order to avoid misunderstandings to readers, we do not directly change the ID in the manuscript, but express the meaning of the ID in a clearer way. For example, SY is replaced by the soil sample plot developed from limestone, which is convenient for readers to understand directly and avoid excessive use of the sample ID.
Comments 8: Results: In all tables and Figures, the order of treatments varies. Please use the same order for the sites in all Tables and Figures.
Response 8: Thank you for your detailed comments. We have added tables and changed pictures in the manuscript, and re-labeled them. At the same time, the numbers of tables and pictures mentioned in the manuscript have also been modified.
Comments 9: Line 244 and others: Consider significant digits in your quantitative data. It is hard to believe that the measurements were that accurate; even if that were the case, it would be distracting and below the significant differences between treatments.
Response 9: Thank you very much for your suggestion. We have modified the decimal part of the parameter in the manuscript so that it was consistent with the expression of the table.
Comments 10: Lines 264-266: That section is redundant and should be suppressed.
Response 10: We have deleted this part of the redundant information, thank you for your professional guidance
Comments 11: Figure 3: Identify altitudes with values or numbers such as 1-2 to 100-200 m, 2-3 to 200-300m, and so forth.
Response 11: Thank you for your advice. We have taken into account the overall quality of the figure and the expression of information, the image and comments have been updated and modified to make it more clear and beautiful, and changed the figure annotations in the entire article.
Comments 12: Line 324: The term “held reliability” sounds odd. Does it mean “is reliable”?
Response 12: We have modified the statement. Thank you very much for your detailed suggestion.
Comments 13: Line 366: Separate “is”
Response 13: Thank you very much for your suggestions and detailed reviews. We have modified the analysis and statement expression of this part.
Comments 14: Discussion: Section 5.1 discusses the occurrence of Glomus, but Paraglomus had many OTUs, which should be addressed.
Response 14: Thank you for your advice. We have referred to the previous research on the distribution of AMF in karst areas. The results showed that the Glomus group in karst areas has unique advantages, and the flora has unique adaptability and characteristics. In addition, our study found that the number of species identified by Glomus was far more than that of Paraglomus, and the distribution of Glomus was related to lithology. Finally, we comprehensively considered the size of the article occupied by this part. Therefore, we focused on the relevant information and flora distribution characteristics of Glomus, hoping to give readers a reference for the study of mycorrhizal fungi in different lithological areas of karst.
Comments 15: Lines 445-446: This sentence is redundant.
Response 15: We have deleted this part of the redundant information, thank you for your suggestion.
Comments 16: Line 463: Replace suggesting with suggested.
Response 16: We have changed the writing of this word. Thank you for your detailed comments.
Comments 17: Lines 545-556: Since there were no significant differences in the diversity index, this section is speculative and should be removed, as it is not based on the data the authors had.
Response 17: Thank you for your suggestion. After reading the manuscript carefully, we found that this part is not related to the manuscript data, and we agree with your point of view. We have deleted this part of the content, and supplemented the diversity index table and the difference analysis results in the manuscript.
Comments 18: Lines 561-568: This section is redundant.
Response 18: We have deleted this part of the content and only retained the meaningful analysis part related to the data. Thank you for your suggestion.
Comments 19: Line 633: The authors state that the rhizosphere significantly correlates with several variables. It must be specified which rhizosphere traits correlate with those variables.
Response 19: Thank you for your detailed comments, we are very sorry that the statement is wrong, the correct meaning is that the content of SOM, AN and ACa in the rhizosphere soil of S. japonica is significantly positively correlated. We have modified the statement and we are very sorry to cause misunderstanding.
Reviewer 2 Report
This study focuses on evaluating the influence of altitude and lithology on the diversity and composition of arbuscular mycorrhizal fungi (AMF) in the rhizosphere of Sophora japonica growing in karst areas of Guangxi, China. The results of this study are of interest for enhancing our understanding of the complex interactions between AMF and medicinal plants in karst ecosystems, providing new insights into how these interactions are regulated and mediated and which factors account for its composition and structure. This contributes to a better prediction of their significance and utilization for ecological restoration and sustainable cultivation practices.
However, there are some drawbacks on the manuscript presentation, which must be improved. Firstly, the absence of access to the sequencing data prevents us from evaluating the accuracy of the analyses. Secondly, the absence of the number corresponding to each representative OTU for every sample in the abundance table complicates data interpretation for readers. Additionally, relying solely on a small portion of the SSU for species identification, especially in AMF, may not provide sufficient certainty in distinguishing between different OTUs. In my view, the authors should employ the OTU number instead of the name of the best match from the sequence found in the MaarjAM database. The results of statistical analyses should be provided in the supplementary materials to support statements made in the text.
Some suggestions and comments have been added directly to the attached document. Please review the comments and suggestions provided within this document.

Author Response
Dear reviewer, I express sincere thanks to you on behalf of all the authors. We are very fortunate to meet professional, responsible, patient and careful experts like you. This is very important for the improvement of our manuscript quality. It also broadens our academic thinking and enhances our ability to express. We seriously considered the reviewer 's suggestions, and revised the article abstract, introduction, method, pictures, tables, result description and so on. The English expression, unit symbols, method reference and details, picture beauty and clarity of the manuscript have been greatly improved. Thank you again for the professional guidance, thank you for your sincere help.
Comments 1: AMF in S. japonica rhizosphere.
Response 1: Thank you for your suggestion, we have modified the statement.
Comments 2: please indication or explain this better.
Response 2: Thank you for your detailed comment, we have modified the expression to avoid misunderstanding for readers.
Comments 3: this means that each composite sample was constituted by three roots and rhizosphere soil. Does this mean that only one sample was taken for each condition? Please clarify.
Response 3: Thank you for your question, we are very sorry to cause misunderstanding. Three replicate plots were set for each plot. The sample of each replicate plot was a biological composite sample formed by random combination of three root and rhizosphere soil samples. We have modified the sampling method in the manuscript.
Comments 4: ???
Response 4: We have deleted this space
Comments 5: The plant roots were cleaned with a brush before collecting the S. japonica rhizosphere soil, which was subsequently divided into two parts.
Response 5: Thank you very much for your advice, we are very sorry, this is our mistake. We have modified the statement
Comments 6: initial denaturation
Response 6: Thanks for your careful guidance, we have re-described the PCR experimental process.
Comments 7: followed by 30 cycles of: denaturation at 94 °C for 30 seconds, annealing at 52 °C for 30 seconds, and extension at 72 °C for 30 seconds. This was followed by a final extension step at 72 °C for 10 minutes. The extracted DNA was then stored at 4 °C for preservation.
Response 7: Thank you very much for your suggestion, which makes us not too simplified in describing the PCR experimental process, but also convenient for readers to understand.
Comments 8: amplions
Response 8: Here refers to the recovery of PCR amplification products, we have rewritten this part.
Comments 9: please indicate their means first
Response 9: Thank you for your advice. We have marked the full name of each abbreviation in the detection of soil physicochemical properties in the materials and methods section.
Comments 10: please format the table in order to facilitate its reading
Response 10: Thank you for your advice. We have made comprehensive changes to the existing tables and new tables, including fonts, font sizes, annotations, column numbers, etc., to ensure the readability and completeness of the tables.
Comments 11: significant differences
Response 11: Thank you for your suggestion, we have modified the annotation.
Comments 12: community
Response 12: We have modified the sentence. Thank you for your help.
Comments 13: spore per gram
Response 13: Thank you very much for your suggestion, we have modified the unit of spore density, which makes our expression more standardized.
Comments 14: Please indicate the altitudes in each soil. Additionally, please add statistical data to the plot; otherwise, indicate in the legend that there were no differences.
Response 14: We redrawn the picture of the colonization and added the results of the significance analysis, along with annotations to help readers understand. At the same time, the altitude information of each number was described in detail in the picture annotation. Thank you very much for your suggestions.
Comments 15: Please indicate the altitudes in each soil. Additionally, please add statistical data to the plot; otherwise, indicate in the legend that there were no differences.
Response 15: Thank you for your suggestion, we redrawn the picture of spore density, marked the results of significance analysis on the picture, and annotated it.
Comments 16: as we do not have access to this data, a supplementary table should be provided with the distributed number os reads per OTU across al samples.
Response 16: Thank you for your suggestion. We provide the number of data upload in the manuscript, but it is not public for the time being. We can provide the results of sequencing and the corresponding statistical table.
Comments 17: Can this be due to the fact that the OTU represented only one sequenced is being considered? Rare sequences should be discarded and normalization of data must be done prior to statistical analysis...
Response 17: Thank you for your suggestion. These data were based on the table of sequencing results. These OTUs were identified to the class level. We count these data mainly to reflect the proportion of each AMF genus in the soil fungal community. The data table will be provided in a supplementary form.
Comments 18: in that supplementary table can be add the best match in each OTU.
Response 18: Thank you for your suggestions, we will add a table covering the distribution of AMF OTUs and the level of identification of each sample.
Comments 19: replace this table by a relative abundance graph
Response 19: We deleted the table and updated the relative abundance figure in the manuscript, while adding a rarefaction curve to reflect the relationship between the amount of sequencing and species diversity. Thank you for your suggestion.
Comments 20: please add these results in a table with the statistical results
Response 20: Thanks to your suggestion, we have added a table of species diversity indices (Richness, Chao1, Shannon, Simpson indices) to the manuscript and added the results of significance analysis.
Comments 21: please redo the NMDS demonstrating that altitude and lithology influenced the composition of the AMF community
Response 21: Thanks for your suggestion, we re-analyzed the NMDS of AMF communities in the rhizosphere of S. japonica under different lithology and altitude treatments, updated the stress parameters and the p values of AMOVA and ANOSIM, and modified the expression in the manuscript.
Comments 22: again, indicate the results in a table that can be added as supplementary data to support this statement
Response 22: We have expressed the updated AMOVA and ANOSIM results in the text and provided them in the form of supplementary materials. Thank you for your professional advice.
Comments 23: include this text in the Figure 5 legend.
Response 23: Thank you for your advice, we are very sorry this is our mistake, we have caused omissions in the processing of manuscript format template, we have included this part of the content to the figure annotations
Comments 24: again, it is mandatory to show all related statistical analysis results to prove this statement. The RDA per se do not indicate the significant differences but is a method to extract and summarise the variation in a set of response variables that can be explained by a set of explanatory variables. The authors can use anova to verify significance of your RDA
Response 24: Thanks for your professional guidance, we have fully considered the analysis methods and language expressions in the manuscript, and found that there were indeed omissions in the analysis. Therefore, we have replaced the analysis method and changed the description expression of this part in the manuscript.
Comments 25: with such a high proportion, let's say the majority, of undetermined factors are responsible for the composition of AMF communities, will it make sense to mention that the variables analyzed here are important for the AMF composition?
Response 25: Thank you for your question. This has caused us to think deeply. The results of VPA reflected the explained proportion of microbial community structure changes of specified environmental factors to a certain extent. The related research on lithology and altitude is generally carried out on a large scale, and the low interpretation rate here may be limited by the scope of the study. The detected environmental factors cannot cover all the influencing factors of the ecosystem, and there are other ecological factors that affect the microbial community. However, the environmental factors detected in this study do have their own significance, and also provide reference for subsequent research. We also combine the results of CCA and VPA to analyze. In the future, we will expand the scope and depth of research on the investigation of microbial community drivers. Thank you again for your suggestions.
Comments 26: .
Response 26: Thank you for your detailed advice, we have modified the symbol
Comments 27: belonging to
Response 27: Thanks for your detailed comments, we have modified the statement and the analysis results of this part.
Comments 28: not all variable contribute significantly to the communities differences, please indicate the statistical data that indicates significant factors responsible for communities composition...
Response 28: Thank you for your suggestion, we have replaced RDA with CCA and modified the results of the analysis.
Comments 29: please add this text to the Figure 7 legend
Response 29: We 've included this section in the comments, and we 've checked and revised the full text. Thank you for your detailed review.
Comments 30: same here. please add this to the legend
Response 30: Thank you very much for your advice, we are very sorry that this is our omission, we have made changes.
Comments 31: these names should be replaced by OTU number...
Response 31: Dear reviewer, thank you for your advice. All the authors attached great importance to this suggestion, and discussed and carefully thought about it, and communicated with the sequencing company. Here are our thoughts. In high-throughput sequencing, the relationship between OTU and species has always been a concern. OTUs are artificially clustered according to 97 % similarity. There are two common situations, that is, multiple OTUs correspond to one species and multiple species correspond to one OTU. The genome of a specie may contain multiple copies, and the similarity between copies may be less than 97 %, which is divided into different OTUs, which should be regarded as the same specie. The division of different strains into the same OTU is very common in 16S analysis. Our results showed that there were multiple OTUs corresponding to one species. In the correlation analysis, OTUs can be analyzed by abundance merging, and ordinates can also be mapped using species. The latter method we think readers may be more intuitive understanding of relevance. Therefore, after comprehensive consideration, we have revised the language expression at the correspondence analysis of the manuscript to avoid difficulty for readers to understand. In the supplementary content of the manuscript, we will also provide the table of species annotation and relative abundance for readers to download. We are very fortunate to meet a responsible reviewer like you. We sincerely thank the reviewer for the understanding and help.
Comments 32: please add this to the legend
Response 32: We have included this section in the annotation, thank you for your guidance.
Comments 33: was verified
Response 33: We have rephrased this paragraph and thank you for your guidance.
Comments 34: please add this to the legend of Figure 10
Response 34: Thank you for your suggestion, we have included this part of the content into the figure 's annotations to avoid misunderstandings by readers.
Comments 35: Characteristics of the AMF community in the rhizosphere of S. japonica
Response 35: We have modified the title of the paragraph and checked and modified the full text. Thank you very much for your suggestion.
Comments 36: Glomus has also been identified as a predominant genus in the rhizosphere of other medicinal plants, including Rosa laevigata Michx.
Response 36: Thank you very much for your guidance to the sentences, which is of great help to improve the quality of our manuscript English expression.
Comments 37: ed
Response 37: We have modified the word, thank you very much for your detailed comments.
Round 2
Reviewer 2 Report
I thank the authors for considering the previous suggestions and making the abundance table available so that the data is visible to all readers. Although I understand the authors' arguments regarding the difficulty in affiliating different OTUs to different classes, genera and species within the Glomeromycota phylum, I also understand that encompassing similar OTUs (at least at a taxonomic level) facilitates the overall analysis. However, not all OTUs even within the same group have the same relevance from the point of view of the plant's performance role. That said, correlations and all statistical analyzes in general, in my opinion, should be done using different OTUs even if in taxonomic terms they represent identical organisms if not the same…
Examining the supplementary table reveals the detection of 5767 OTUs. Among these, 2628 OTUs, constituting a mere 0.085% of the total, exhibit fewer than 10 sequences per OTU. Consequently, these OTUs are considered rare or are likely attributable to sequencing errors. Did the authors take this into consideration during their analysis?
Upon reviewing the taxonomy associated with the OTUs (presumably derived from the Maarjam database), it becomes evident that only a small fraction of the representative sequences is identified at the genus and species level. Hence, Table 2 lacks coherence and fails to accurately represent the obtained data. Consequently, it should be removed. The abundance and distribution are effectively illustrated in Figure 4. Furthermore, it corroborates my previous statement regarding the limited identification of OTUs at the genus level. Indeed, based on this figure, it's evident that Glomus is not the most abundant, contrary to what is stated in the abstract and text. This prompts the question of whether the subsequent analyses presented in the manuscript were conducted by grouping OTUs based on affiliation, or if they were carried out considering only specific OTUs, or even all OTUs were analyzed?
Abstract
L17-18- As this result is derived from only 270 OTUs, it may not adequately represent the diversity of AMF present in the rhizosphere soil. Therefore, I suggest that the authors specify the diversity in terms of OTUs to provide a more comprehensive understanding.
M&M
L171 My apologies for the mistake. Please replace 'amplion' with 'amplicons'. Thank you for your understanding.
Results
L254-258 This sentence inadequately represents the results. Figure 1 illustrates a notably high rate of AMF colonization (>70%), with no significant differences observed between the various lithologies and altitudes, except for sandstone at 171.45m, which exhibits a significantly lower AMF colonization rate. On the contrary, Figure 2 indicates a clear influence of lithology on spore density. Specifically, spore density is notably higher in dolomite soils compared to limestone and sandstone soils, which exhibit similar densities. Nevertheless, limestone soils at an altitude of 537.40m displayed a spore density comparable to that observed in dolomite soils.
Please verify if the SY1 and SY2 statistics are accurate. According to the image, these soils display a spore density similar to limestone, which appears to contradict the statistical information presented in the figure.
L277-283 please rephrase this sentence. The provided data inadequately represents the full spectrum of diversity observed, encompassing less than 5% of the total. Notably, only OTU_1 exhibits an abundance exceeding 11%, which is double that of all other mentioned OTUs. This underscores the necessity for authors to present data on each OTU, rather than solely focusing on those associated with identified species or genera.
Table 2-delete it since it does not represent the full spectrum of diversity found
Figure 3- present this figure as supplementary data
Figure 10. Could you confirm whether this figure includes all OTUs or solely those associated with identified species or genera? If it's the latter, then the analysis should be reconsidered, focusing at least on the most abundant OTUs rather than solely on those with limited representation.
Figure 11- Same question as above for figure 10.
Author Response
Dear reviewers, we feel very happy and very lucky to get your professional guidance again. On behalf of all the authors, I express our most sincere thanks to you. Thank you for your detailed guidance on manuscript English expression, pictures, tables, results, abstract, etc. According to the reviewer 's suggestions, we further revised the manuscript, including tables, pictures, language, supplementary materials and so on. We believe that with your help, the expression of the manuscript has been greatly improved, and at the same time, it also presents readers with higher manuscript quality. Thank you again for your sincere help and understanding.
Comments 1: Abstract L17-18- As this result is derived from only 270 OTUs, it may not adequately represent the diversity of AMF present in the rhizosphere soil. Therefore, I suggest that the authors specify the diversity in terms of OTUs to provide a more comprehensive understanding.
Response 1: Thank you for your suggestion, we supplemented the number of sequences and OTUs, as well as the identified species in the abstract to facilitate readers to fully understand the diversity and species annotation results of the rhizosphere microbial community analysis of S. japonica.
Comments 2: M&M L171 My apologies for the mistake. Please replace 'amplion' with 'amplicons'. Thank you for your understanding.
Response 2: Thank you for your advice. We understand your concerns very well. We have modified the word.
Comments 3: Results L254-258 This sentence inadequately represents the results. Figure 1 illustrates a notably high rate of AMF colonization (>70%), with no significant differences observed between the various lithologies and altitudes, except for sandstone at 171.45m, which exhibits a significantly lower AMF colonization rate. On the contrary, Figure 2 indicates a clear influence of lithology on spore density. Specifically, spore density is notably higher in dolomite soils compared to limestone and sandstone soils, which exhibit similar densities. Nevertheless, limestone soils at an altitude of 537.40m displayed a spore density comparable to that observed in dolomite soils.
Response 3: Dear reviewer, thank you for your advice, we are very sorry that we did not describe the results in detail, and thank you again for your demonstration of the language expression of the manuscript. According to your suggestion, we re-expressed the description of colonization and spore density in the manuscript. Thank you very much for your sincere help.
Comments 4: Please verify if the SY1 and SY2 statistics are accurate. According to the image, these soils display a spore density similar to limestone, which appears to contradict the statistical information presented in the figure.
Response 4: Dear reviewer, thank you for your advice, we checked the colonization and spore density data, the analysis of the figure is correct. We did not mix all the data together for one-way ANOVA and LSD, and the Duncan multiple test. Although the mixed data may be more significant, it does not reflect the importance of group sampling, and it is difficult to see the differences between lithology and altitude. It will also appear that the data analysis is more chaotic. Here is the analysis of three different groups, but because the letters are all lowercase, may have caused a misunderstanding, we are very sorry. The two groups were tested by two independent-sample t-tests (such as SY1 and SY2). The three groups were tested by one-way ANOVA and LSD, and the Duncan multiple test (p < 0.05) (such as BY1, BY2, BY3; SH1, SH2, SH3, SH4, SH5). In order to facilitate readers ' understanding, we have supplemented the analysis method in the data analysis section. We sincerely thank you for your suggestions.
Comments 5: L277-283 please rephrase this sentence. The provided data inadequately represents the full spectrum of diversity observed, encompassing less than 5% of the total. Notably, only OTU_1 exhibits an abundance exceeding 11%, which is double that of all other mentioned OTUs. This underscores the necessity for authors to present data on each OTU, rather than solely focusing on those associated with identified species or genera.
Response 5: Dear reviewer, we agree with your suggestion very much. When we update the relative abundance picture, we also re-described the distribution of microorganisms in the manuscript. Thank you very much for your suggestion, which makes us understand the characteristics of microbial community distribution in many aspects.
Comments 6: Table 2-delete it since it does not represent the full spectrum of diversity found.
Response 6: Table 2 only represents the current retrieval results of our test data in the database, which cannot fully prove the diversity of AMF in plant rhizosphere soil. We just want to give readers a reference. Thank you for your suggestion, we have deleted Table 2.
Comments 7: Figure 3- present this figure as supplementary data.
Response 7: Thank you for your suggestion, we have put Figure 3 into the supplementary file.
Comments 8: Figure 10. Could you confirm whether this figure includes all OTUs or solely those associated with identified species or genera? If it's the latter, then the analysis should be reconsidered, focusing at least on the most abundant OTUs rather than solely on those with limited representation.
Response 8: Dear reviewer, we thank you for your suggestions. In the analysis of the figure, we compare all OTUs with the database to obtain species annotation information, and analyze the correlation of species. After the agreement of all authors, we decided to add the correlation analysis between the top 50 OTUs and environmental factors in the manuscript. We are very grateful for the reviewer 's advice so that the manuscript can meet different readers ' understanding of OTUs and species.
Comments 9: Figure 11- Same question as above for figure 10.
Response 9: Dear reviewers, we sincerely thank you for your advice and help. We carefully consider the expression and intention of figure 11, here is mainly to use the results of correlation analysis and Mantel test to further express the relationship between environmental factors and microbial community diversity richness. We understand the concerns of reviewer very much. According to the reviewer 's suggestion, after discussion by all the authors, we believe that OTU can be achieved here. However, after referring to many articles, we believe that it may be more appropriate to use the abundance of microbial genera here. There are three main reasons : first, it is convenient for readers to understand ; second, it is beautiful to draw ; third, if readers want to understand the correlation between OTUs and environmental factors, we also provide the results of correlation analysis between species and OTU and environmental factors in Figure 9. In order to avoid duplication and redundancy of information, the authors believe that it is appropriate to use the abundance of the genus level for analysis. On behalf of all the authors, I would like to express sincere thanks to the reviewer for your patient guidance and help.
Round 3
Reviewer 2 Report
I appreciate the authors' changes and responses. In fact, the scientific quality of the manuscript is greatly improved. I'm just reticent about figure 9 in the new version of the manuscript, where there are two Spearman correlation images analyzing the relationship between environmental factors and the top 50 OTUs and another with the identified AMF species. My problem is with the second image that of species. In fact, if we look at the horizontal lines on the right side I cannot identify AMF species. What is “sp.”? “Wirsel_OTU16”? Etc. This affiliation does not correspond to species. If the authors insist that it is important to make this correlation, then they will have to be rigorous and present the name of the species in the figure. It cannot go like this for publication as it does not help readers at all, on the contrary, it confuses them.
Figure 9- Please review the name of the AMF species shown in the image. If you cannot improve and present the names correctly, please delete the figure.
Author Response
Dear reviewer, we sincerely thank you for your professional and patient help. We are very sorry that the review of the manuscript took a lot of your time and energy. At the same time, we are very lucky to get your guidance three times. According to your suggestions, we have re-searched the species information and drawn the figure, so as to ensure the quality of the manuscript and present the readers with clearer figure. We are very grateful to the reviewer for the guidance, which gives us a new understanding of the professionalism of scientific research and manuscript writing. Once again to express our most sincere thanks to the reviewer.
Comments 1: Figure 9- Please review the name of the AMF species shown in the image. If you cannot improve and present the names correctly, please delete the figure.
Response 1: Dear reviewer, thank you for your advice. We re-inquiry the AMF species information in the MaarjAM database (https://maarjam.ut.ee/), and can find the classification information of each species at the species level on the figure. We thought and recreated the figure added the genus-level names of the species to the figure and annotation, and explained the meaning of sp and unassigned. In fact, the name of the AMF species involved in the figure can be found in Table 2 (deleted), and the OTU table provided in the supplementary file of our manuscript can also find the corresponding species information. Thank you for your suggestions, which makes us rethink the expression of figure. While improving the clarity of figure, it also avoids readers' doubts and meets readers' diversified reading needs. We sincerely thank you for your guidance again.
